# Breast Imaging Physics in Mammography (Part I)

**DOI:** 10.3390/diagnostics13203227

**Published:** 2023-10-17

**Authors:** Noemi Fico, Graziella Di Grezia, Vincenzo Cuccurullo, Antonio Alessandro Helliot Salvia, Aniello Iacomino, Antonella Sciarra, Gianluca Gatta

**Affiliations:** 1Department of Physics Ettore Pancini, Università di Napoli Federico II, 80126 Naples, Italy; 2Radiology Division, ASL Avellino, 83100 Avellino, Italy; graziella.digrezia@gmail.com; 3Nuclear Medicine Unit, Department of Precision Medicine, Università della Campania Luigi Vanvitelli, 81100 Napoli, Italy; vincenzo.cuccurullo@unicampania.it; 4Department of Precision Medicine, Università della Campania Luigi Vanvitelli, 81100 Napoli, Italy; antoniosalvia89@gmail.com (A.A.H.S.); gianluca.gatta@unicampania.it (G.G.); 5Department of Human Science, Guglielmo Marconi University, 00193 Rome, Italy; nelloiacomino@libero.it; 6Department of Experimental Medicine, University of Campania Luigi Vanvitelli, 80138 Napoli, Italy; antonella.sciarra@unicampania.it

**Keywords:** breast imaging, medical physics, mammography, breast cancer

## Abstract

Breast cancer is the most frequently diagnosed neoplasm in women in Italy. There are several risk factors, but thanks to screening and increased awareness, most breast cancers are diagnosed at an early stage when surgical treatment can most often be conservative and the adopted therapy is more effective. Regular screening is essential but advanced technology is needed to achieve quality diagnoses. Mammography is the gold standard for early detection of breast cancer. It is a specialized technique for detecting breast cancer and, thus, distinguishing normal tissue from cancerous breast tissue. Mammography techniques are based on physical principles: through the proper use of X-rays, the structures of different tissues can be observed. This first part of the paper attempts to explain the physical principles used in mammography. In particular, we will see how a mammogram is composed and what physical principles are used to obtain diagnostic images.

## 1. Introduction

Breast cancer is the most frequently diagnosed cancer in women in Italy [1]. Risk factors include age, reproductive factors, hormonal factors, dietary and metabolic factors, previous radiotherapy in the chest, previous breast dysplasia or neoplasia, familiarity and heredity [2,3,4]. Thanks to screening [5] and increased awareness among women, most breast malignancies are diagnosed at an early stage when surgical treatment can more often be conservative and the therapy adopted is more effective [6,7,8]. Mammography is a specialised radiological technique [9] for analysing and diagnosing breast-related diseases, in particular breast cancer [10,11,12]. The need to develop a specialised clinical technique, breast imaging, arose because of the high incidence of cancer in the female population: in the West, about 1 in 8 women (12%) have the possibility of developing breast cancer during their lifetime [13]. In recent years, there has been a slight increase in the incidence of breast cancer, probably due to a greater extension of diagnostic investigations and population screening that leads to the detection of more cases, often early, than in the past [14]. Periodic screenings are essential for early detection of cancer and specialised instruments need advanced technology to achieve quality diagnoses [5]. Mammography is the gold standard for early detection of breast cancer. Its low cost, low radiation dose to the patient and high sensitivity, i.e., the ability to detect cancer in positive subjects, make mammography the best diagnostic technique [15,16,17].

The development of mammography began in the 1930s, but the evolution of detection techniques and the use of new materials have made it possible in recent years to achieve high levels of sensitivity and specificity for the diagnosis of breast cancer [18,19,20]. In particular, with the support of new techniques such as radiomics, [21] tools from artificial intelligence [22] have been implemented for staging or tumour diagnostics to achieve more accurate diagnoses [23,24]. Mammography has a high specificity (>85), i.e., it allows a patient who is actually free of breast cancer to be declared negative with a high reliability [25]. The diagnosis is based on the search for possible masses in the patient’s tissue [26]. The masses may be benign or malignant tumours. Once the mass lesion has been operated on, histological examination will confirm the diagnosis [27]. The considerable complexity of the anatomy of the breast and the uniqueness of its anatomical details makes it extremely complex to produce a diagnosis [28]. Each patient has a particular structure of the glandular tree and, in particular, the tumour mass appears as a distortion of the glandular tree, hence the texture [29]. Clinical experience is the discriminating factor in a reliable diagnosis of breast disease [30,31,32,33,34,35,36,37,38] (Figure 1).

In terms of X-ray attenuation, there are small differences between normal breast tissues and those in which tumour masses are present: X-ray attenuation in glandular tissue is close to that of adipose tissue. [39] Since the attenuation coefficient of the medium depends on the energy of the incident radiation, it can be observed that, for the same thickness of the crossed medium, a differential attenuation of the tissue is greater at low energies (10–15 keV) and lower at high energies (>35 keV) [40,41,42]. This makes it necessary to use special equipment designed to optimise the diagnosis of breast tumours [43,44,45]. The spatial resolutions required in a diagnostic mammographic examination are high since there may be microcalcifications and or a cluster of microcalcifications, which may be an indicator of the presence of a tumour. The detection of microcalcifications, however, requires a high resolution in the 50 μm range, which is not achievable with ordinary general radiographic equipment with very small focal spots (0.3 mm) [46,47,48,49].

From an anatomical point of view, the breast is composed of a sac formed by 80% adipose tissue bound to the sac by means of appropriate Cooper’s ligaments and fibroglandular connective tissue (gland). In particular, the glandular fraction is defined as the percentage weight fraction of the gland divided by the total weight of the breast (gland and adipose tissue inside the pouch) [50]. The focus on the gland is greater than that on the adipose or vascular tissue. The glandular fraction is relevant because a breast with a high glandular fraction has a higher radio-opacity (ability to take X-rays) and is more likely to have a tumour, as well as complicating diagnosis considerably (high mammographic density). The image shows the linear attenuation trend as a function of the beam energy, and a marked difference can be seen between the attenuation relative to the different adipose and glandular tissues [51,52,53]. This difference is more pronounced at low energies, due to higher differential contrast, and becomes more subtle at higher energies, due to lower differential contrast. The problem with mammographic imaging is the almost overlapping attenuation curves for glandular tissue or (IDC) infiltrated ductal carcinoma [54,55,56,57,58,59].

## 2. The Mammograph

Differentiated X-ray equipment is needed for specialised mammographic investigations as a high resolution is required; this has led to the development of specialised medical imaging to localise, for example, microcalcifications, which are often only a few tens of microns in size. They require very small focal spots, about 0.3 mm; the negligible difference between the attenuation in healthy and tumour tissues requires the use of energy spectra of 15 or 18 keV [60,61,62,63].

An apparatus structured in this way is a mammograph (Figure 2) and consists of:Radiogeni tubes and detectors,Compression devices,Anti-diffusion grids,Automatic exposure meters.

Compared to X-ray imaging equipment, a mammograph is equipped with a motorised compression plate, generally made of polycarbonate, which can be up to 100 N. Compression is necessary to reduce the amount of dose administered and to obtain high quality images: by reducing the thickness through which the radiation passes, the number of scattered photons reaching the detector also decreases (Figure 2) [64,65,66].

The operator ensures that the breast is compressed and moved away from the pectoralis major muscle by craniocaudal (CC) Figure 3 and medio-lateral-oblique (MLO) Figure 1 compression; this exposure can make any tumour masses more obvious as they are often located close to the muscle pectoral (∼20%) (Figure 3 and Table 1).

### 2.1. Cathode and Filament 

In the focusing cup, mammography devices have two filaments that produce focal spots with nominal sizes of 0.3 mm and 0.1 mm. Focal spots of 0.1 mm are used for in- magnification mammography, where the high spatial resolutions required are achieved by minimising geometric blurring. A non-linear relationship is established between the current in the filament and the current in the tube due to space charge effects that are modulated by appropriate feedback circuits which regulate the current in the filament, such that the desired current in the tube is achieved, which is approximately 100 ± 25 mA for filaments of focal spot 0.3 mm and 25 ± 10 mA for filaments of focal spot 0.1 mm [67].

### 2.2. Anode 

Anode Mammography systems use rotating anodes generally made of molybdenum, rhodium or tungsten. A SID (distance between the source and the image) of about 65 cm leads to a cut off of the field of view generally equal to 24 cm × 30 cm; for this reason, the tube and anode are inclined and the sum of the inclinations constitutes the anode angle (effective angle of the anode can be up to 24 degrees, but generally is 0–16° anode, 8–24° tube). The electronic focal spot is larger than the optical focal spot, which is why small anode angles provide a good thermal load as the tube current increases. The central axis, homologous to the central ray for general radiography, does not bisect the beam. For both configurations, A and B in the figure, we observe that the beam has a direction parallel to the patient’s chest wall due to the effective inclination of the anode. Starting from the focal spot and cutting orthogonally to the plane of the reflector, the correct radiation exposure of the anatomy concerned is ensured [68].

### 2.3. Heel Effect 

The Heel effect is a phenomenon that occurs in general radiography and mammography apparatus and concerns the variation in beam intensity along the cathode–anode direction. As you move away from the cathode and towards the anode, there is a decrease in the intensity of the radiation beam reaching the detector. The anatomy of the breast follows a decreasing trend of intensity in terms of thickness, so the device is designed to place the patient on the side of the central axis. A lower beam intensity implies fewer photons reaching the detector; so, under these conditions, a uniform thickness could produce significant image degradation on the angle side; it is possible to balance the Heel effect by exploiting the difference in thickness between the chest junction and the nipple [69].

### 2.4. Focal Spot

High spatial resolutions are required in mammography and this is achieved by using small focal spots and, therefore, small anode angles to limit geometric blurring. For magnification mammograms, the range of focal spots is 0.1 mm to 0.15 mm, while for contact mammograms the sizes are 0.3 mm to 0.4 mm. Higher SIDs reduce the magnification and, thus, the influence of the focal spot on the image resolution; so, if the SID is set at 65 cm, it is necessary to use focal spots of at least 0.3 mm, and if the SID is increased, larger focal spots can be used. In mammography, the focal spot falls on the edge of the detector and the central axis falls on the side closest to the patient’s chest wall. The bisector of the radiation field cone is called the reference axis and is used to obtain the size of the optical focal spot. Different from what happens in a general X-ray apparatus, the central axis and the reference axis do not coincide because of the inclination of the effective anode angle (θ), but, by knowing the geometrical parameters, it is possible to trace the dimensions of the optical focal spot at any point of the detector. Taking ϕ as the reference angle between the anode and the reference axis, it is possible to evaluate the length of the optical focal spot (MFL); the length of the optical focal spot at the reference axis is linked to that at the central axis by the law:(1)$$MFL_{reference-axis}=MFL_{central-axis}\left(1-\frac{tan\left(\theta -\phi \right)}{tan\vartheta }\right)$$

The length of the optical focal spot is maximum at the central axis (ϕ − → θ: sotto al catodo) and varies along the cathode–anode direction (Figure 4).

## 3. Radiation Beam: Quality, Filters, Collimation 

### 3.1. Beam Quality and Filtration

Models and software simulations suggest the optimal trade off to deliver the lowest dose to the patient with the highest contrast. This is achieved for a single-energy X-ray beam with variable energy between 15 keV and 25 keV, depending on the patient’s anatomy, particularly considering the thickness of the tissue passed through and the glandular fraction. Experimentally, it is observed that the spectrum of the X-rays emitted by the tube is a polyenergetic spectrum, in which the low energy component contributes substantially to the dose administered to the patient and contributes less to the quality of the image. At high energies, the dose to the patient decreases, as does the contrast. A compromise has to be reached by using appropriate combinations of target material on the anode and filtration. In this way, it is possible to generate characteristic X-ray radiation (XRF) with a behaviour very similar to the ideal mono-energy spectrum identified by computational simulations (17–23 keV). Typically, the materials of which the anode of a radiogenic tube is made up are Molybdenum _42_Mo with characteristic peaks at $E_{L\to K}^{K\alpha }=17.5\;keV$ and $E_{M\to K}^{K_{\beta }}=19.6\;keV$, $Rhodium\;{(}_{45}Rh)$ with characteristic peaks at $E_{L\to K}^{K_{\alpha }}=20.2\;keV$ and $E_{M\to K}^{K_{\beta }}=22.7\;keV$ and, recently, $Tungsten\;{(}_{74}W)$.

To ensure good structural integrity, together with a low attenuation, the tube port uses a beryllium window ${(}_{4}Be)$ with a thickness of just 1 mm: the tube’s inherent filtration must be kept low to ensure the transmission of the photons that carry the diagnostic information. By applying other filters of the same material as the target on the anode, it is possible to favour the preferential transmission of photons with the energies of the characteristic lines (XRF), removing the photons of the beam that belong to the regions of the spectrum with very low and very high energies. Generally, additional filters used in mammography for Molybdenum targets are made of Molybdenum with a thickness of 0.03 mm, or for Rhodium targets are made of Rhodium with a thickness of 0.025 mm. They contribute to the removal of those photons in the beam whose energies contribute to the dose to the patient but not to the image quality [11,18,26,27]. Combinations of the above target/filter systems are also used, such as the Mo/Rh combination, but the Rh/Mo combination cannot be used as an additional Molybdenum filter would completely attenuate the discrete spectrum (XRF) of the Rhodium target [70,71,72].

### 3.2. Collimation and Field of View

Within a mammography apparatus, the beam is collimated with appropriate actuators. The field of view framed depends on the specific diagnostic examination: small fields are referred to if they are 18 cm × 24 cm in size. Collimation is optically guided to ensure correct anatomical exposure. Variable geometry shutters make it possible to avoid the possibility that the breast occupies only a small portion of the field of view (small anatomical dimensions), resulting in a degradation of image quality along the edges of the breast projected onto the receiver. In the area just adjacent, in fact, the maximum number of photons would arrive, transmitted due to the absence of absorbing medium, negatively affecting the quality of the image at the contour [73,74,75,76].

### 3.3. Generator and Automatic Exposure Control System (AEC)

The generator is a further component of the mammograph, which differs from those used in general radiology in the voltage values used. There is a compensation circuit for the space charge effect which alters the linearity of the relationship between the filament current and tube current [44,77,78,79,80] (Figure 5).

Overall, the AEC enables the application of the optimal radiographic technique. It can be seen from the figure that the sensor of the automatic exposure meter is located underneath the detector and is responsible for automatic exposure control. It is a device that exploits the information provided by the compression plate and, through appropriate algorithms, starts a test export lasting 100 ms to assess the density of the breast. After adjusting the density by means of a selector switch and setting the duration of exposure, the exposure meter in full automatic mode is able to modulate the peak voltage kVp, additional filtration and type of target on the anode if the exposure test suggests a better performance with a target of a different material [11,62,81,82].

## 4. Compression: Contrast and Dose

### 4.1. Compression

The mammography technique is characterised by the use of a motorised plate that triggers compression of the patient’s breast. The thickness obtained depends on the type of diagnostic examination and also depends on the glandular action. Less dense breasts have a ∼20% glandular fraction while denser breasts have a ∼50%, glandular fraction, which affects the energy spectrum required with consequent intervention of the AEC in the choice of target material on the anode. Compression is also useful in reducing the overlap of anatomical structures. The different mechanical stiffness of different tissues causes a lateral displacement so that a tumour tissue can be more easily distinguished from a healthy glandular tissue.

The plate is capable of exerting a force that can be modulated in a range from 10 N to 30 N. Isovolumic compression allows the thickness to be reduced to an overall size between 9 cm and 2 cm:Less X-ray scattering: they pass through a smaller thickness and due to compression and slippage of the different tissues, there is less overlap of the tissues interacting with matter with a lower probability of Compton events so the tract is in fact shorter, which results in a lower SPR through compression;Less image degradation: compression brings the tissue closer to the detector plane by reducing magnification, so details of anatomical structures are more evident and by lowering the quality of the diagnostic information produced, geometric blurring is avoided: f = (M − 1)F;Lower absorbed dose: breast tissue absorbs less radiation as the number of photons removed from the beam by absorption of the material depends on the thickness of the material (at the same linear attenuation coefficient), in accordance with the Lambert–Beer law.

The choice of plate thickness is crucial: a large plate thickness would increase mechanical rigidity and, therefore, resistance to deflection, but would produce greater beam attenuation on the patient’s surface. The compression plate cannot be deflected, it must be flat. Deflections of up to 1 cm in the compression area are tolerated, but larger values would affect the image quality. The adjustment of the compression can be given by the operator or by the automatic exposure system (AEC) and depends on the anatomical breast characteristics to be exposed; commonly used force values range from 10 N to 20 N. Compression, with the same anatomical characteristics, results in a significant reduction in SPR.

### 4.2. Contrast

The radiation transmitted by breasts consists of: -Primary photons: the primary radiation ‘carries’ the diagnostic information regarding the attenuation of the different breast tissues and provides maximum contrast of the subject. -Scattered photons: scattered radiation is only a part of the total radiation and is an additive contribution that degrades the contrast of the subject. 

Defining the maximum subject contrast, the one that would be obtained in the absence of diffuse radiation, as
(2)$$C_{0}=\frac{\Delta P}{P}$$
with $\frac{\Delta P}{P}con\frac{\Delta P}{P}$, is the difference between the signal in an ROI and the signal in the background, divided by the background. Then, the maximum contrast in the presence of diffuse radiation is given by
(3)$$C_{s}=\frac{C_{0}}{1+SPR}\mathrm{SS}$$
where $SPR=\frac{S}{P}$ S is the scatter to primary ratio, where S is the number of photons diffuse, while P is the number of primary photons. The dependence between the maximum contrast in the presence of diffuse radiation and the SPR justifies the firm compression of the breast: a reduction in thickness leads to a significant decrease in the SPR ratio, resulting in an increase in image contrast.

The SPR ratio also depends on the glandular fraction. In mammography, scattered radiation increases with breast thickness and diameter, but no appreciable change is observed as a function of kVp. The range in which the mammograph operates generates photons of such energy that the preferred interaction mechanism is photoelectric, rather than Compton scattering events. Therefore, there are no significant variations in the number of photons scattered, with peak voltage variations in a range from 25 kVp to 35 kVp.

$SPR=\frac{S}{P}$ is scatter to primary ratio, where S is the number of diffuse photons, while P is the number of primary photons. The techniques used to reduce scattered radiation include the use of antiscatter grids and the possible interposition of air between the patient and the detector, a technique known as air gaps. The most frequently used anti-scatter grids have a one-dimensional structure with parallel lines and a low grid ratio of 4:1 or 5:1, compared to that used in general radiography. The covering material is usually made of aluminium or carbon fibre. The spatial frequencies depend on the type of grid installed in the bucky potter that contains the detector, the grids and the AEC sensor. They range from 30 lp/cm to 50 lp/cm for moving grids, and the high frequency oscillation allows the position of the shadow of the spectra projected onto the detector to be varied continuously, thus equalising the loss of information. In comparison, fixed grids can reach up to 80 lp/cm. 

### 4.3. Magnification

Mammography magnification is achieved by using the filament on the cathode, which produces the focal spot with a nominal size of 0.1 mm. Subsequently, a support platform is installed for the udder at a fixed distance above the detector (OID). Then, the anti-scatter grid is removed and, finally, the compression plate is used. Magnification mammography allows greater separation of anatomical structures at the expense of image quality due to geometric blurring. In general, the main advantages of magnification mammography are a higher resolution of anatomical structures on the image detector by a factor equal to the magnification factor and a reduction in noise and scattered radiation. Disadvantages include geometric blurring, caused by the finite size of the focal spot, as well as the limiting operating conditions of the apparatus: a small focal spot constrains the current that can circulate in the tube, resulting in a lengthening of the exposure time required for the diagnostic examination; however, long exposures result in low quality images [37,48,83].

### 4.4. Dosimetry in Mammography 

In mammography, quality controls are carried out to assess the dose delivered and the image quality. In Europe, there is no directive on mammographic patient dose limits for individual diagnostic examinations, but the protocol has a limit of around 2.5 mGy. With good optimisation, it is possible to perform a mammography examination well below this limit, i.e., 1.5 mGy to 2.0 mGy. Quality controls are performed on dummies and their purpose is to verify that the apparatus is operating within the required quality standards without exceeding the reference dose limits. The graph shows the mean absorbed glandular dose (mrad) as a function of the thickness of a compressed breast for three peak voltage values. The average glandular dose is the average dose absorbed by the entire glandular tissue. It is not possible to know in advance how the gland is distributed within the breast tissue: if it were located more near the upper surface at the entrance to the beam, there would be more radiation deposition. If the glands were closer to the lower surface, close to the detector, the absorbed dose would be much lower due to the attenuation of the tissues above. Using a Monte-Carlo simulation, it is then possible to obtain an estimate of the average dose to the gland by considering the total energy absorbed by the whole gland divided by the mass of the gland itself. The information provided by the simulation relates only to glandular tissue as adipose tissue cannot develop tumours. A scatter plot obtained from the experimental data of the patients examined allows a retrospective investigation to verify the performance of each individual mammography device [84,85,86].

### 4.5. Average Glandular Dose

Compression of the breast reduces the dose to the patient: with the same amount of material passed through, and therefore the same linear coefficient μ, a negative change in thickness leads to greater transmission of radiation and, therefore, less energy deposition in the medium passed through, according to the Lambert–Beer law. The Lambert–Beer law describes the phenomenon of attenuation of a photon beam, where attenuation is defined as the phenomenon of removal by absorption or scattering from the direction of propagation by the crossed medium:(4)$$N=N_{0}e^{-\mu x}$$

Consider an ionisation chamber located at the entrance to the patient surface, below the compression plate, which allows the exposure to be measured, the subsequent conversion of the Roentgen measurement value to mGray then provides an estimate of the Air Kerma at the patient entrance. 

This measurement is carried out in the absence of the breast by placing the ionisation chamber where the patient’s surface comes into contact with the compression plate, since the radiation backscattered by the exposed anatomy would alter the value provided by the ionisation chamber.

If we now set a new level, for example, at a depth x with respect to the upper surface of the patient’s anatomy, then the intensity of the radiation undergoes an attenuation, which can be described by the Lambert–Beer law; similarly, it is the case for Kerma, with a law that decreases exponentially:(5)$$K=IAKe^{-\mu x}$$

Assuming we want to measure the dose value at a certain depth x since this depends on the Kerma and the thickness of the material through which the radiation passes. For this purpose, the Dose or collision Kerma is defined as
(6)$$D=\Psi {\left(\frac{\mu_{en}}{\rho_{0}}\right)}_{E}$$
and where ${\left(\frac{\mu_{en}}{\rho_{0}}\right)}_{E}$ is the evaluated mass energy absorption coefficient for energy E, Ψ is the energy fluence of a mono-energy beam, given by the product of the photon fluence and its energy, and is measured in $\left[keV\cdot cm^{-2}\right];\left(SI:\left[J\cdot m^{-2}\right]\right)$:(7)$$\Psi =\left(\frac{Fotoni}{Superficie}\right)\cdot \left(\frac{Energia}{Fotoni}\right)=\Phi \cdot E$$

On the other hand, while the mass energy absorption coefficient does not depend on the thickness crossed but only on the type of tissue and the energy E, the energy fluence Ψ will decrease with the thickness of the material crossed by the radiation, contributing to the exponential reduction in the dose being the Ψ function of the photon fluence, which is related to the thickness of the material crossed by an exponential attenuation law. The compression of the breast results in a reduction in the dose to the breast tissue as this reduces the thickness of the anatomy through which the radiation passes; the reduction in thickness means less attenuation of the photon fluence so more photons are transmitted per unit area and less energy is deposited in the tissue, thus there is a lower dose. However, since the distribution of glandular tissue within the breast is not first known, we can estimate the optimal dose amount by Monte-Carlo simulations, in which the total energy absorbed by a certain type of breast and for a certain type of beam is estimated, and the ratio of this value to the estimated gland mass allows us to define a fundamental mammographic dosimetric parameter, such as the mean glandular dose (MGD or AGD). Quantitatively, in mammography, the mean glandular dose (MGD or AGD) is calculated as follows: the IAK is measured using an ionisation chamber placed under the compression plate in the absence of the breast and multiplied by a coefficient called the normalised glandular dose coefficient (DGN), according to
(8)$$D_{MGD}\left[mGy\right]=DGN\left[\frac{mGy}{mGy}\right]\cdot K_{IAK}\left[mGy\right]$$

The normalised glandular dose coefficient (DGN) is calculated using the MonteCarlo simulation for a certain mammographic spectrum (e.g., 28 kVp Mo/Mo) and for certain anatomical characteristics of the breast. The mean glandular dose (MGD or AGD) is reported as searchable information in the header lines of the mammogram DICOM [87,88,89,90,91,92,93].

## 5. Conclusions

In this first part of the paper, we looked in detail at the physical operation of a mammogram. Radiology has indeed had to develop a specialized technique for breasts given the complex anatomy that is highly variable for each patient. We looked at how the structure of a mammograph is composed and how much this setting can affect the emission and detection of X-rays and the resulting image obtained. In particular, we observed the importance of the quality of the beam; it is polyenergetic, thus the low energy component affects the image quality by lowering it and contributes to the dose administered, so you have to induce a trade off to try to optimize the administered dose and obtain the best possible image. We intervene in this way on the beam quality. Collimation, but also the compression of the breast, plays a fundamental role, particularly to disfavour the presence of scatter. In this paper, in conclusion, we analysed the operation of the mammograph from a physical point of view, observing critical points and strengths. We have seen that it is important to make improvements to this technology as it is used as a key screening technique for the prevention and early detection of common cancers and, therefore, is useful in lowering the mortality incidence. We will see in the following work the difference between mammography and more advanced diagnostic techniques and where technological development aims to go in the near future.

## Figures and Tables

**Figure 1 diagnostics-13-03227-f001:**
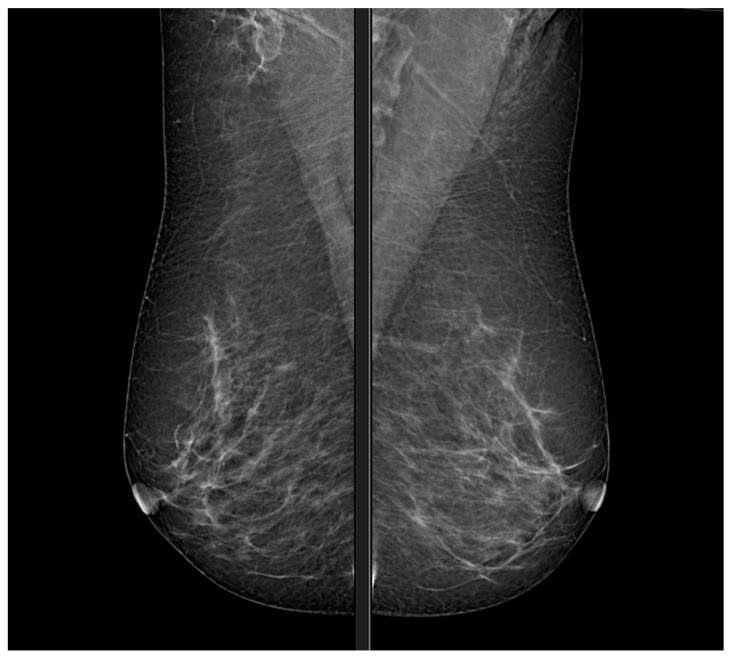
Mammography, MLO view.

**Figure 2 diagnostics-13-03227-f002:**
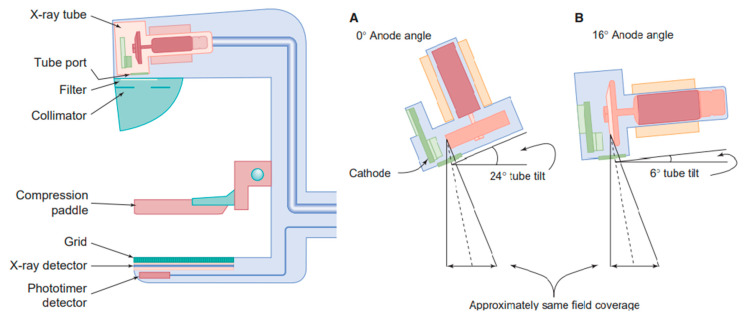
Left, a dedicated mammography system, right, anode angle in mammography system, an anode angle of 0 (**A**) and 16 degrees (**B**), require a tube tilt of 24 and 6 degrees [44].

**Figure 3 diagnostics-13-03227-f003:**
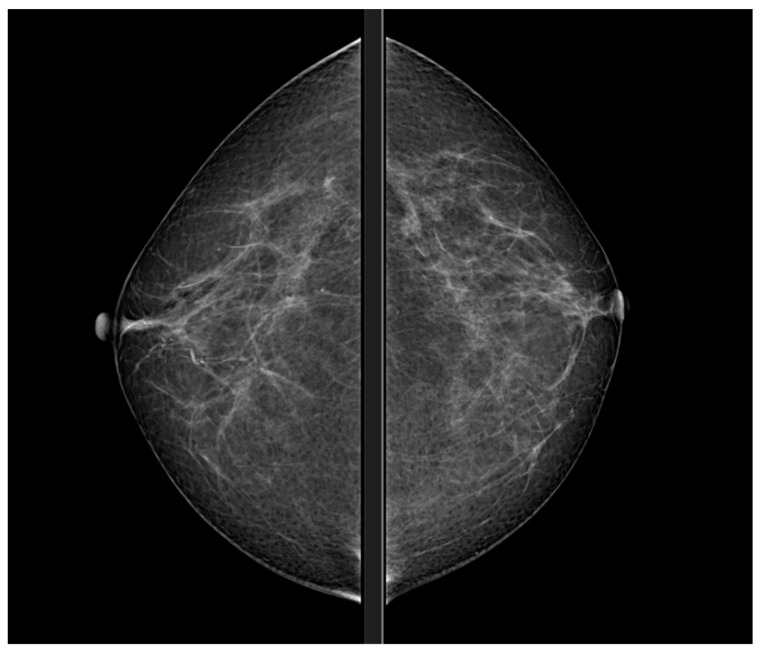
Mammography, CC view.

**Figure 4 diagnostics-13-03227-f004:**
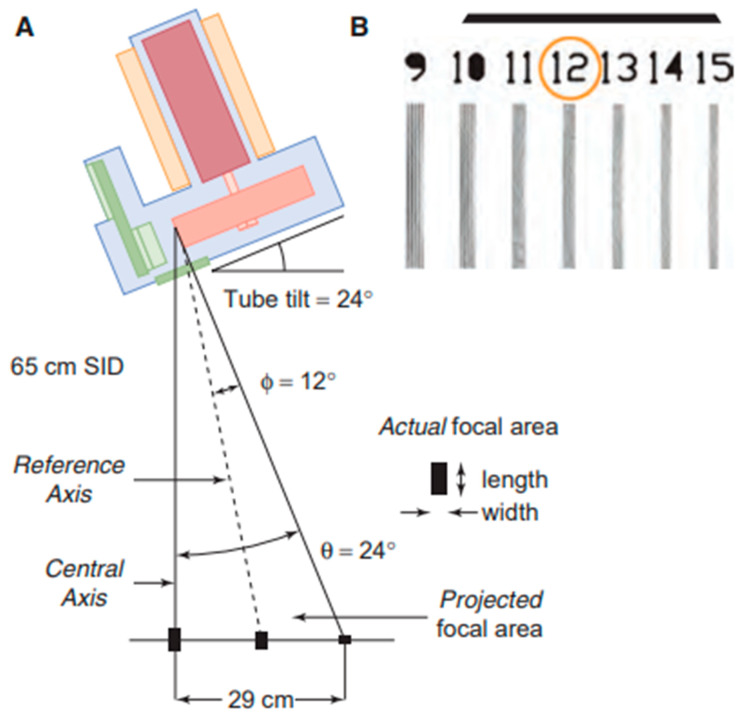
(**A**) the projected focal spot size varies along the cathode–anode axis, the dashed line correspond at 12 degrees. (**B**) a resolution bar pattern with object magnification measures overall system resolution, including that of the focal spot.

**Figure 5 diagnostics-13-03227-f005:**
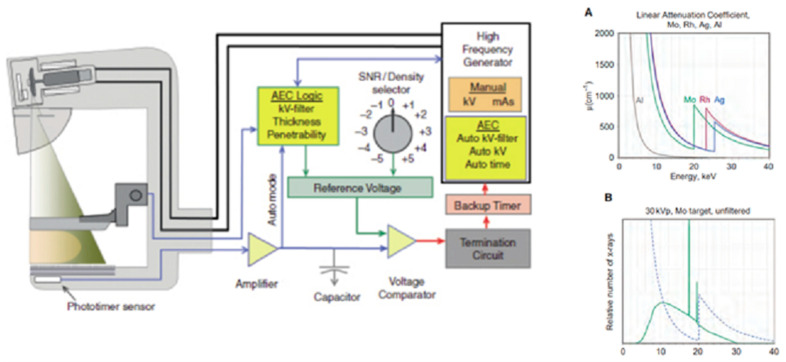
(**A**) linear attenuation coefficient of different target/filter. (**B**) overlap Molybdemun’s spectra (piloted beam at 30 kVp) with the functional trend of the attenuation coefficient (up). Overview of the apparatus and automatic exposure system (AEC) (down) [44].

**Table 1 diagnostics-13-03227-t001:** Nominal focal spot size and measured tolerance limit of mammography X-ray tubes specified by NEMA and MQSA regulations.

Nominal Focal Spot Size (mm)	Width (mm)	Length (mm)
0.10	0.15	0.15
0.15	0.23	0.23
0.20	0.30	0.30
0.30	0.45	0.65
0.45	0.60	0.85
0.60	0.90	1.30

## Data Availability

The data presented in this study are available on request from the corresponding author. The data are not publicly available because are propriety of Università della Campania “Luigi Vanvitelli”, Napoli, Italy.

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
