# Peer review of "Breast Imaging Physics in Mammography (Part I)"

_diagnostics, 2023, doi:10.3390/diagnostics13203227_

Round 1

Reviewer 1 Report (Previous Reviewer 1)

As a diagnostic method considered to be stable, it has been some time since there has been a review article on the technique and principles of mammography. This article is helpful for colleagues who need an in-depth understanding of mammography techniques.

OK

Reviewer 2 Report (New Reviewer)

This is an excellent manuscript that provides a comprehensive overview of the physical principles of mammography. The authors have done a great job of explaining complex concepts clearly and concisely. The manuscript is well-written and well-organized, and it is evident that the authors have a deep understanding of the subject matter.

I particularly liked the authors' discussion of the different types of mammography modalities and their advantages and disadvantages. This will be helpful for readers who are new to mammography or who are considering different options for breast cancer screening. This is a well-written and informative manuscript that is essential reading for anyone interested in the physical principles of mammography. I highly recommend it for publication.

This manuscript is a resubmission of an earlier submission. The following is a list of the peer review reports and author responses from that submission.

Round 1

Reviewer 1 Report

I found that this article is marked as an "Article", and the content of the article is more like a Review about mammography technology. The article introduces in detail the past technology of mammography and some modern issues, such as CEM, breast CT and so on. However, as a review article, this article has too few relevant references and the details are not deep enough. Might be of limited help to colleagues who want to use this article for a comprehensive review or for learning.

The imaging principles of modern technologies (such as breast CT/CEM) and the difficulties encountered (such as recombination of images and overcoming displacement distortion, clinical practicability, etc.) should be discussed in depth.

Reviewer 2 Report

Is a review article on methods for diagnosing breast cancer. It must be recognized that, given the immense amount of data that exists on the various diagnostic techniques for breast cancer, it is extremely difficult to carry out a review. Unfortunately this paper does not meet any of the requirements necessary for a good review.  A review paper is not a simple list of the technical characteristics of the systems currently in use, a critical approach is absolutely necessary, such as a complete view on the most recent bibliography. There is no trace of this in the presented text which discusses the different techniques in an almost scholastic way. Even the title is somewhat misleading as it seems to want to deal with the physical aspects of the different techniques, while merely providing technical information on the equipment used. It seems that the present paper is the first of a series, however it does not say what are the overall aims to be achieved.

The text has several errors and the language is very incomplete.

The bibliography is very limited and the self-citations are very numerous (16 self-citations out of 26 entries). The conclusions are completely missing, in which possibly there should be a perspective vision that is completely missing.

In conclusion, the work cannot be published in this form.